# Study on the Changes in Immobilized Petroleum–Degrading Bacteria Beads in a Continuous Bioreactor Related to Physicochemical Performance, Degradation Ability, and Microbial Community

**DOI:** 10.3390/ijerph191811348

**Published:** 2022-09-09

**Authors:** Yixuan Liu, Weisi Li, Yanlu Qiao, Fangying Yu, Bowen Wang, Jianliang Xue, Mianmian Wang, Qing Jiang, Zhibin Zhou

**Affiliations:** 1College of Safety and Environmental Engineering, Shandong University of Science and Technology, Qingdao 266590, China; 2Shandong Provincial Eco-Environmental Monitoring Center, Jinan 250102, China; 3Institute of Yellow River Delta Earth Surface Processes and Ecological Integrity, Shandong University of Science and Technology, Qingdao 266590, China; 4College of Chemical Engineering and Environment, Weifang University of Science and Technology, Weifang 262700, China; 5Qingdao Port, Qingdao 266500, China

**Keywords:** bioreactor, immobilization, microbial community, bioremediation, oil contamination

## Abstract

Continuous bioreactors for petroleum degradation and the effect factors of these bioreactors have rarely been mentioned in studies. In addition, indigenous bacteria living in seawater could influence the performance of continuous bioreactors with respect to petroleum degradation in practice. In this paper, a bioreactor fitted with immobilized petroleum–degrading bacteria beads was designed for further research. The results indicated that the diesel degradation rate of the bioreactor could remain above 50% over 27 days, while degradation performance decreased with bioremediation time. Intriguingly, the diameters of immobilized petroleum–degrading bacteria beads were reduced by 32.49% after 45 days remediation compared with the initial size of the immobilized petroleum–degrading bacteria beads. Change in immobilized petroleum–degrading bacteria beads was considered to correlate remarkably with reduced degradation efficiency. Therefore, this paper will be helpful for further study and improvement of bioreactors in the practical context of oil-spill accident recovery.

## 1. Introduction

High demand for fuel, such as petroleum, has led to an increase in the number of petroleum-spill accidents. It has been estimated that more than 120 million liters of oil per year enter marine environments [1,2]. When oil-spill accidents happen, a lot of crude oil enters the sea in a short time, causing heavy destruction to the environment and ecological systems [3,4]. Crude oil is mainly composed of hydrocarbons, and it has been found that certain microorganisms may use hydrocarbons as a carbon source [5,6]. However, since there are not sufficient quantities of free petroleum–degrading bacteria in seawater, their ideal remediation effects cannot be achieved [7]. Therefore, it is urgent to find a more efficient approach to remediate petroleum pollution in the sea.

Immobilization technology has been at the forefront of bioremediation research and could be used to improve bacterial density and enhance the degradation of petroleum-spill pollution [8,9,10,11]. According to the research results, immobilized petroleum–degrading bacteria have great advantages in restoring petroleum-contaminated seawater [12,13]. Although numerous studies on immobilized petroleum–degrading bacteria beads have been reported, no continuous-running bioreactor system incorporating petroleum–degrading bacteria beads has been established [14,15,16]. Hence, a bioreactor fitted with immobilized petroleum–degrading bacteria beads was established as a continuous bioreactor in order to further improve the density of petroleum–degrading bacteria.

Researchers have elucidated the degradation mechanism of immobilized petroleum–degrading bacteria beads, which is related to bacterial degradation [7]. The research has demonstrated that the degradation process can be divided into two main parts. In the early stage of degradation, surface adsorption is dominant, whereas biodegradation is dominant in later stages. Diesel was adsorbed by immobilized petroleum–degrading bacteria beads and then was decomposed by microorganisms into small molecular substances, which were then slowly degraded into CO_2_ and H_2_O over time. Moreover, previous studies have proved that bacterial community structure is related to immobilized microbial remediating effectiveness [17]. However, it has only been demonstrated that indigenous bacteria could influence bacterial structure in immobilized petroleum–degrading bacteria beads [18], and the structural changes in microbial communities follow a pattern which has not yet been discovered. In the meantime, the mechanisms related to microorganisms in seawater and immobilized petroleum–degrading bacteria beads are not understood. In addition, there have been only a few studies on the changes in microbial communities, especially on the influence of such changes on continuous remediation in bioreactors. Therefore, further study of the migration and evolution of microbial communities is necessary to explore the bioremediation mechanism.

In this paper, a continual bioreactor was designed to investigate degradation performance. However, during the experiment, the degradation rate declined over time. At the same time, an interesting phenomenon attracted our attention, namely, that the volume of immobilized petroleum–degrading bacteria beads in the bioreactor reduced. In order to further enhance the effectiveness of bioreactors, this phenomenon should be investigated further, as it has only rarely been mentioned in earlier studies. A larger range of available detection methods were used to study and discuss this phenomenon in this research.

## 2. Materials and Methods

### 2.1. Preparation of Immobilized Bacteria

The composition of mineral salt medium (MSM) is displayed in Appendix A, the pH of which should be kept at 7.2–7.5. The petroleum–degrading strain sp1 (Shewanella algae), selected from a previous study [17], was inoculated with 100 mL MSM medium and enriched with 1 mL diesel oil for 3 days. Enteromorpha were heated at 260 °C for 2 h to prepare biochar, a detailed procedure for which was introduced in a previous study [17]. A bacterial culture, fully stirred with Enteromorpha biochar, was added into sodium alginate solution to obtain a mixture. Then, the mixed solution was added to CaCl_2_ solution dropwise to obtain immobilized petroleum–degrading bacteria beads of Shewanella algae, which were stored in CaCl_2_ solution for 72 h. Once they were completely formed, the immobilized petroleum–degrading bacteria beads were taken out and washed with normal saline and distilled water. All solutions mentioned in the experiment were sterilized and cooled to room temperature for use. A specific preparation procedure is shown in Appendix A.

### 2.2. Bioreactor Set-Up and Experimental Design

With an inner diameter, length, and operating volume of 30 mm, 1000 mm, and 0.71 L, respectively, a uniform cylindrical reactor was set up for the experiment. The reactor operated continuously and steadily for 45 days. The entire experiment was run in the bioreactor, and diesel-contaminated seawater was prepared by adding diesel to seawater and stirring fully to ensure the influent diesel concentration. Synthetic diesel-contaminated seawater was pumped in from the leftmost inlet of the unit and out of the rightmost outlet. The bioreactor was equipped with sampling inlet and outlet valves, and the influents and effluents from the device were sampled once a day to measure diesel concentrations for calculation of the degradation rate. The specific design of the bioreactor is shown in Appendix A. Furthermore, the actual seawater used in the continuous experimental period was collected from the Tangdao Bay of Qingdao City, Shandong Province, China. The target influent diesel concentration was controlled at about 1% for the continuous experimental period. 

### 2.3. Analysis Method

The diesel in the water inlet and outlet were extracted with n-hexane in advance and then the concentrations of diesel were determined according to the ultraviolet spectrophotometry method. The degradation rates of diesel in the bioreactor were calculated using the following equation [13]:D_R_ = (C_0_ − C_1_)/C_0_ × 100%
where D_R_ (%) is the degradation rate of diesel, C_0_ (mL/L) is the influent concentration of diesel, and C_1_ (mL/L) is the effluent concentration of diesel.

The immobilized petroleum–degrading bacteria beads, collected from water inlets and outlets after 45-day bioremediation (named IB and EB, respectively), were sent to Shanghai Biotech Co., Ltd. for high-throughput sequencing, as was original seawater (OS), which was filtered through a 0.22 μm-fiber filter. In this study, DNA samples were extracted using a FastDNA^®^ Spin Kit (MP Biomedicals, Santa Ana, CA, USA). The bacterial primers 338F (ACTCCTACGGGAGGCAGCAG) and 806R (GGACTACHVGGGTWTCTAAT) were used to amplify partial 16S rRNA genes [19]. Detailed sample information is shown in Table 1. Portions of these immobilized petroleum–degrading bacteria beads were freeze-dried for further morphological observation and group-typing by scanning electron microscopy (SEM) and Fourier transform infrared spectroscopy (FTIR).

### 2.4. Statistical Analysis

R programming (v.3.3.1, Ross Ihaka and Robert Gentleman, Auckland, New Zealand) tools were used to map community composition based on the data table in the tax_summary_a folder. Mothur (version v.1.30.2, Patrick Schloss, Ann Arbor, Washtono County, MI, USA) was used to cluster sequences into distance-based (97% similarity) operational taxonomic units (OTUs). Furthermore, PICRUSt was used to infer gene representation using taxonomic information from 16S rRNA genes sequencing [20]. The data were analyzed using the online Majorbio Cloud Platform (www.majorbio.com, accessed on 16 March 2022) [21].

## 3. Results and Discussion

### 3.1. Diesel Degradation Rate in a Bioreactor under Continuous Operation

In order to display the changes in degradation effects in the reactor, the influent concentrations and effluent concentrations of diesel were measured daily. During the 45-day remediation of the bioreactor, the influent concentration of diesel was kept at 10 mL L^−1^ approximately, while the effluent concentration of diesel changed on different days. As shown in Figure 1, the varying trend of the degradation rate (depicted as the black dotted line) decreased obviously over time. The highest diesel degradation rate for the bioreactor was calculated as 57.19%, while the lowest value was 37.45% after running for 40 days. Meanwhile, a noteworthy phenomenon occurred in association with the decline in the diesel degradation rate. It was recorded by the photochemical method that the initial reactor was filled fully with immobilized petroleum–degrading bacteria beads at the start of the experiment, while around only two-thirds of the reactor volume was filled with the immobilized beads after running for 45 days. Hence, it was speculated that the decrease in the diesel degradation rate was linked to the decrease in the volume of beads. The immobilized degrading bacteria beads were analyzed by the physicochemical method and high-throughput sequencing in order to determine the reason for this change.

### 3.2. Physicochemical Analysis of Immobilized Petroleum–Degrading Bacteria Beads under Continuous Operation

#### 3.2.1. Morphological Analysis of Immobilized Petroleum–Degrading Bacteria Beads

Regarding morphological changes to the immobilized petroleum–degrading bacteria beads, diameters were measured and SEM was chosen for microcosmic observation. Twenty immobilized petroleum–degrading bacteria beads for an initial sample and twenty beads for a 45-day sample were selected randomly for measurement (Figure 2a). Compared with the initial immobilized petroleum–degrading bacteria beads, the average size of the beads decreased from 41.55 mm to 28.05 mm after 45 days. As for microcosmic morphology, the structures of the immobilized petroleum–degrading bacteria beads varied significantly over the 45-day remediation (Figure 2b,c). A percentage of the initially immobilized petroleum–degrading bacteria beads had regular hole structures, whereas the 45-day immobilized petroleum–degrading bacteria beads had irregular structures, which might have affected the absorption of diesel [7]. A slight force applied to the immobilized petroleum–degrading bacteria beads could destroy their structure, which might be related to the environment of the immobilized petroleum–degrading bacteria beads.

#### 3.2.2. Group Types and Number of Immobilized Petroleum–Degrading Bacteria Beads

To investigate the change in quantity and in the kinds of groups on the immobilized degrading bacteria beads, FTIR analysis was performed to determine the influence of the continuous process on the immobilized petroleum–degrading bacteria beads in the reactor. Such an analysis was undertaken to study the variation mechanism in the bioreactor with regard to the immobilized degrading bacteria beads. Compared with the petroleum–degrading bacteria beads initially immobilized in the bioremediation reactor, the distribution of the characteristic peaks of the beads had not changed significantly after the period of bioremediation. However, quantitative analysis of the immobilized petroleum–degrading bacteria beads in the bioreactor showed that the strength of the characteristic peaks of the immobilized degrading bacteria beads at 0 and 45 days changed considerably. According to the study of Fanesi et al., the contribution of spectral features to the prediction of the content of each macromolecule was specific, by which protein structure, phosphorylated molecules, and lipid acylchains could be characterized [22,23]. As shown in Table 2, the intensities of the characteristic peaks at 1500–1800 cm^−1^, 1400–1500 cm^−1^, and 900–1200 cm^−1^ related to cell proteins, fatty acids, glycopeptides, and phosphate groups of nucleic acid constituents, respectively, were weaker in the spectra for the immobilized degrading bacteria beads. In contrast, the peaks at 2800–3000 cm^−1^, representing cell membrane fatty acids, were strengthened with respect to the initial immobilized petroleum–degrading bacteria beads [24]. The variation in the typical peaks indicated that microbial composition was different in the initially immobilized petroleum–degrading bacteria beads, in correspondence with the detection results of the high-throughput sequencing [24,25]. As for -OH (stretching) in molecules, the association peak shape was wider at the peak of 3000–3500 cm^−1^, while the association degree was greater, with the peak being wide and the wave number being low. -OH in the sample was evidently lower than in the initial sample in Figure 3. On the one hand, hydrogen-bonding interactions in alginate could contribute to the toughness of the immobilized petroleum–degrading bacteria beads [26]. On the other hand, sodium alginate could interact with diesel through hydrogen bonds to promote the adsorption of diesel components [7]. Thus, a decrease in -OH might induce variation in the immobilized petroleum–degrading bacteria beads and further influence the degradation of the bioreactor.

### 3.3. Community Structure of Immobilized Petroleum–Degrading Bacteria Beads in the Bioreactor

#### 3.3.1. Biological Diversity

Alpha diversity was used in this study to show community richness and community diversity according to the Chao index and the Shannon index, respectively. Community richness and community diversity showed similar varying trends, OS being the highest, as shown in Figure 4. The initial immobilized petroleum–degrading bacteria beads were made with single bacteria (*Shewanella algae*), so that the Shannon and Chao indices were evidently both lower than the values for the immobilized petroleum–degrading bacteria beads after 45 days in the IB and EB. Change in community richness and community diversity was affected by change in environment and bacterial migration. A previous study reported that adjusting the living conditions of the immobilized petroleum–degrading bacteria beads would induce bacterial migration and community structure [17]. However, different sites in the bioreactor showed different community richness and community diversity values for the immobilized petroleum–degrading bacteria beads. The Chao index for the EB (199.62) was close to the index for the IB (217.54), indicating that community richness in the two areas of the bioreactor was similar. Nevertheless, Shannon indices differed between the influent site (2.67) and the effluent site (1.49) of the bioreactor, showing that the immobilized petroleum–degrading bacteria beads at the water inlet had a higher diversity, which may have been linked to the degradation of diesel in the bioreactor. More details related to community change will be introduced in Section 3.3.2.

#### 3.3.2. Community Structure

The community structures in the original seawater (OS) and in the immobilized petroleum–degrading bacteria beads at the water inlet (IB) and at the outlet (EB) showed the abundances for each sample at the family level (Figure 5a). Advantageous bacteria could be obviously analyzed in the different environments, where the family *norank_f__norank_o__SAR202_clade* (14.70%) was the most abundant in the original seawater, and the family *Flavobacteriaceae* had the highest community structure percentages in the EB and IB (78.46% and 51.28%, respectively). In fact, the family *Shewanellaceae* should be the only strain in the initial immobilized petroleum–degrading bacteria beads; however, it accounted for just 1.65% (EB) and 6.00% (IB) after 45 days in the bioreactor. Hence, the detection results indicated that the family *Shewanellaceae* decreased in the immobilized petroleum–degrading bacteria beads significantly, whereas the family *Flavobacteriaceae* increased after continuous remediation, which should be studied more in detail. 

A heatmap was constructed to analyze the change in community composition at the genus level (Figure 5b). Except for the genus *Shewanella*, many bacteria, not being abundant in seawater, could be detected in the immobilized petroleum–degrading bacteria beads, which indicated that the immobilized petroleum–degrading bacteria beads provided shelter for these bacteria and improved their growth and reproduction. The genera *Alcanivorax*, *Shewanella*, *Thalassospira*, *Alteromonas*, *Tenacibaculum*, and *Lutibacter*, were abundant in the interior of the immobilized petroleum–degrading bacteria beads, and these bacteria were all petroleum–degrading bacteria [27,28,29]. Plenty of holes in the immobilized petroleum–degrading bacteria beads could absorb diesel (shown in Figure 2c); thus, the immobilized petroleum–degrading bacteria beads attracted many petroleum–degrading bacteria to migrate into the interior of the immobilized petroleum–degrading bacteria beads [30,31]. However, the genus *Maribacter*, belonging to the family *Flavobacteriaceae*, could directly be observed to be dominant in the immobilized petroleum–degrading bacteria beads in the detection results. Moreover, effluent immobilized petroleum–degrading bacteria beads showed more *Maribacter* than influent immobilized petroleum–degrading bacteria beads.

#### 3.3.3. The Main Difference in the Bacteria between Immobilized Petroleum–Degrading Bacteria Beads

Fisher’ exact test was used to compare species abundance differences between the dominant bacteria in the interiors of the immobilized petroleum–degrading bacteria beads at the inlet and outlet of the bioreactor. The confidence interval for genus *Maribacter* was −35.65 ± 0.64%, showing a significant difference between the two samples in Figure 6. On the one hand, other dominant petroleum–degrading bacteria also showed significant differences between the two samples and mainly existed in the influent immobilized petroleum–degrading bacteria beads. On the other hand, there was a high percentage of the genus *Maribacter* in the immobilized petroleum–degrading bacteria beads at the effluent site, corresponding to the research results.

#### 3.3.4. COG Functional Classification Statistics

Further investigation of functional abundance was carried out in order to determine the influence of continuous remediation on the immobilized petroleum–degrading bacteria. Compared with species composition, COG functional composition was similar across all samples in Figure 7. The analysis showed functional abundance with regard to the transportation and metabolism of amino acids, nucleotides, coenzymes, and lipids were all lower in the immobilized petroleum–degrading bacteria beads than in the seawater. On the contrary, the research indicated that the immobilized petroleum–degrading bacteria beads at the effluent site were the most abundant in terms of function related to carbohydrate transport and metabolism.

### 3.4. Analysis of the Change in the Immobilized Petroleum–Degrading Bacteria in the Bioreactor

According to photochemical and microcosmic observations, the morphologies of the immobilized petroleum–degrading bacteria beads in the bioreactor changed obviously after 45-day continuous bioremediation. Irregular hole structure could not only induce fragility in the immobilized petroleum–degrading bacteria, it could also influence the absorption of diesel, further affecting the degrading functions of the immobilized petroleum–degrading bacteria [7]. In the FTIR detection, variation in the typical peaks at 3800–4000 cm^−1^, 1500–1800 cm^−1^, 1400–1500 cm^−1^, and 900–1200 cm^−1^ indicated changes in microbial composition in the immobilized petroleum–degrading bacteria beads [24,25,26]. Corresponding to the detection results, high-throughput sequencing confirmed the change in microbial community structure in the immobilized petroleum–degrading bacteria beads. Further analysis of community structure showed petroleum–degrading bacteria belonging to the genera *Alcanivorax*, *Thalassospira*, *Alteromonas*, *Tenacibaculum*, and *Lutibacter*, which became abundant in the interiors of the immobilized petroleum–degrading bacteria beads along with the genus *Shewanella* [32,33,34]. Moreover, the most abundant genus, *Maribacter*, was proven to decompose carbohydrates, such as alginate, which fact was linked to the FTIR detection result showing that hydrogen-bonding interactions in alginate decreased [26,35,36]. COG functional composition also verified the abundance of functions in terms of carbohydrate transport and metabolism, which were evidently higher in the later immobilized petroleum–degrading bacteria beads. These findings correspond to previous analysis results showing that the genus *Maribacter* accounted for a higher percentage of community composition in the immobilized petroleum–degrading bacteria beads. Hence, the variation in functional genes in the immobilized petroleum–degrading bacteria beads was related to the bacteria using carbohydrates. Increasing numbers of carbohydrate-degrading bacteria would affect the physical structure of the immobilized petroleum–degrading bacteria beads; thus, when more carbohydrates were being consumed by bacteria, the degradation rate in the bioreactor would have declined over time.

## 4. Conclusions

In order to investigate the decline in degradation performance in the petroleum–degrading bioreactor used for oil-spill accident recovery, microbial population dynamics were analyzed as well as the physicochemical parameters of the immobilized petroleum–degrading bacteria beads after 45 days of continuous operation. Together with the detection results for the bioreactor, the following conclusions can be drawn:

(1)The bioreactor could maintain a degradation rate above 50% during earlier weeks, indicating that renewal of the immobilized petroleum bacteria beads over time would be beneficial for bioreactor performance;(2)Bacterial migration led to more kinds of petroleum–degrading bacteria, such as *Alcanivorax*, *Shewanella*, and *Thalassospira*, and so on, becoming part of the community structure in the immobilized petroleum–degrading bacteria beads; however, the total abundance of petroleum–degrading bacteria decreased significantly;(3)The increase in the representation of the genus *Maribacter*, which consumes alginate, further changed the morphology of the immobilized petroleum–degrading bacteria beads, thus influencing the efficiency of the bioreactor [35,36].

It can be concluded that the genus *Maribacter* was the main reason that the immobilized petroleum–degrading bacteria beads were destroyed and that the performance of the bioreactor decreased further. Thus, it will be necessary to find a way of inhibiting the reproduction of the genus *Maribacter* in the bioreactor in order to improve the performance in the next study.

## Figures and Tables

**Figure 1 ijerph-19-11348-f001:**
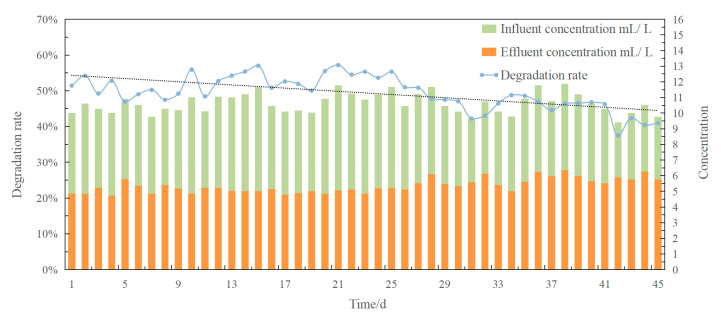
The degradation rate of the immobilized petroleum–degrading bacteria beads in the bioreactor over 45 days. (The influent concentrations and effluent concentrations of diesel are shown in green and yellow, respectively.)

**Figure 2 ijerph-19-11348-f002:**
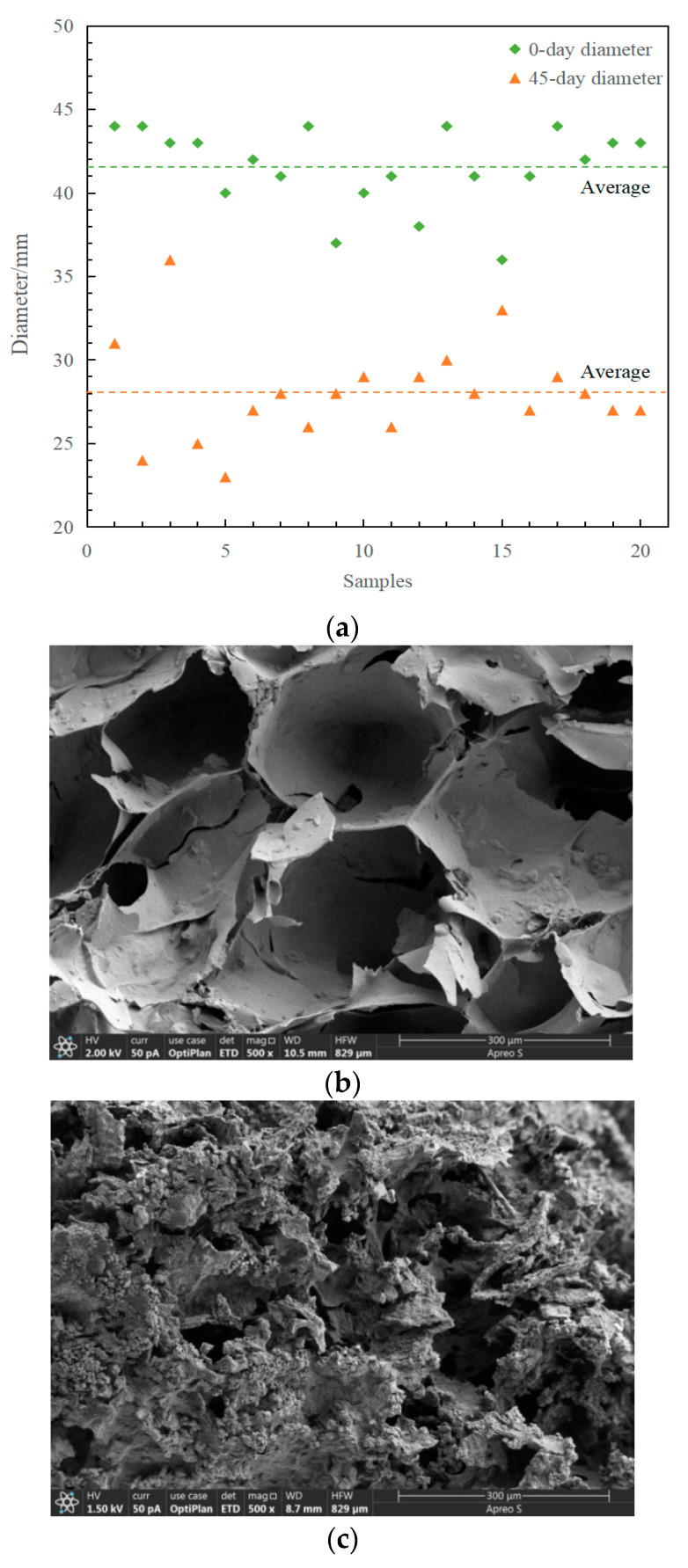
(**a**) Diameters of the initially immobilized beads (◆ in green) and the immobilized beads in the bioreactor after 45 days (▲ in orange). Interior structural changes in (**b**) the initial immobilized beads and (**c**) the immobilized beads in the bioreactor after 45 days recorded by SEM.

**Figure 3 ijerph-19-11348-f003:**
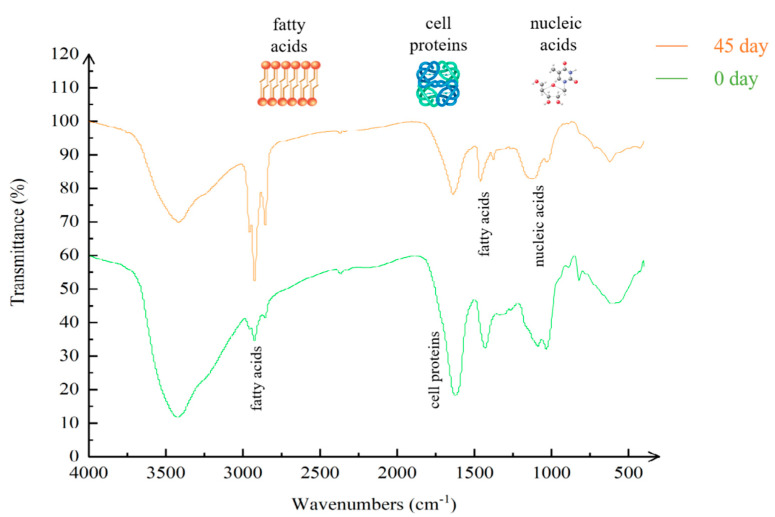
Fourier transform infrared spectroscopy (FTIR) analysis of the initial immobilized petroleum–degrading bacteria beads and the immobilized petroleum–degrading bacteria beads in the bioreactor after 45 days, represented by the green and orange lines, respectively.

**Figure 4 ijerph-19-11348-f004:**
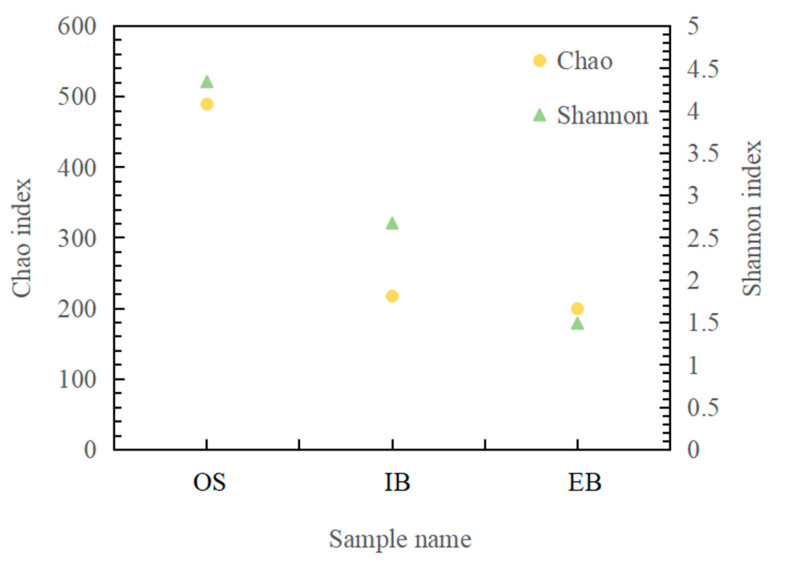
Biological diversity of the original seawater (OS) and the immobilized petroleum–degrading bacteria beads in different areas of the bioreactor (water inlet site (IB); water outlet site (EB)). (Shannon indices are shown as green triangles; Chao indices are shown as yellow circles.)

**Figure 5 ijerph-19-11348-f005:**
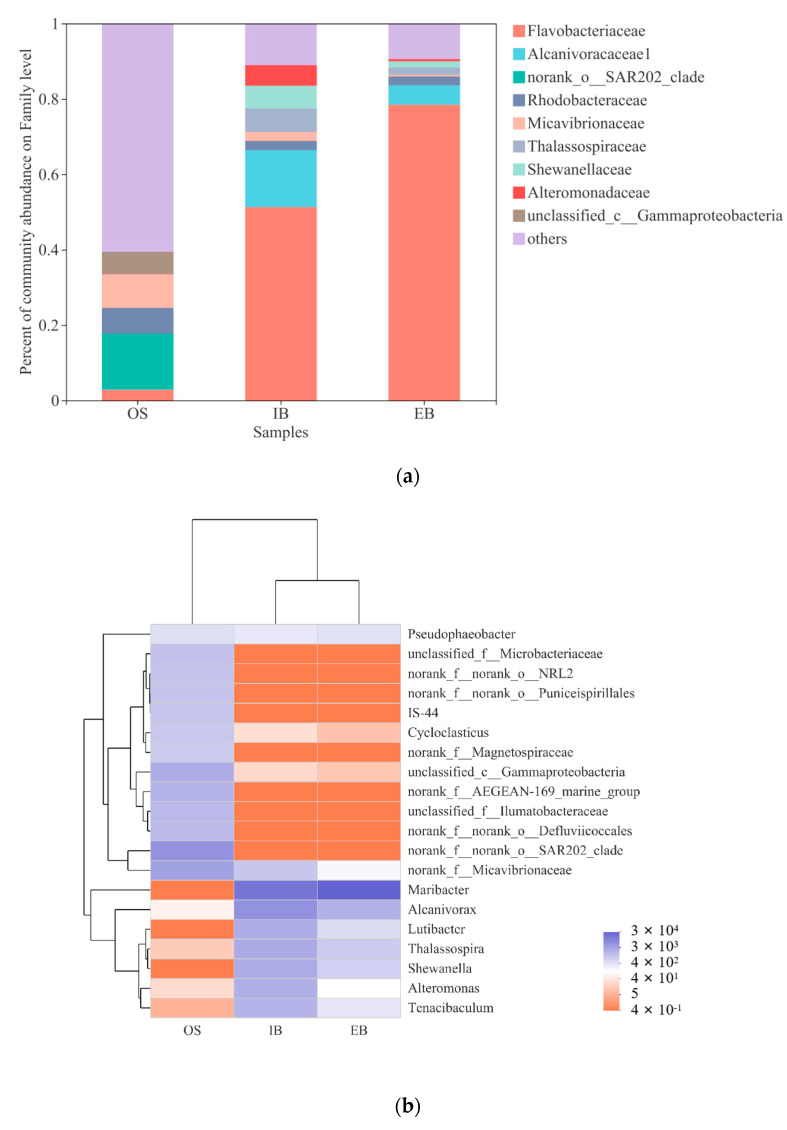
(**a**) Community structures at the family level in the original seawater (OS) and in the immobilized petroleum–degrading bacteria beads in different areas of the bioreactor. (**b**) Community composition heatmap at the genus level for the original seawater (OS) and for the immobilized petroleum–degrading bacteria beads in different area of the bioreactor (water inlet site (IB); water outlet site (EB)).

**Figure 6 ijerph-19-11348-f006:**
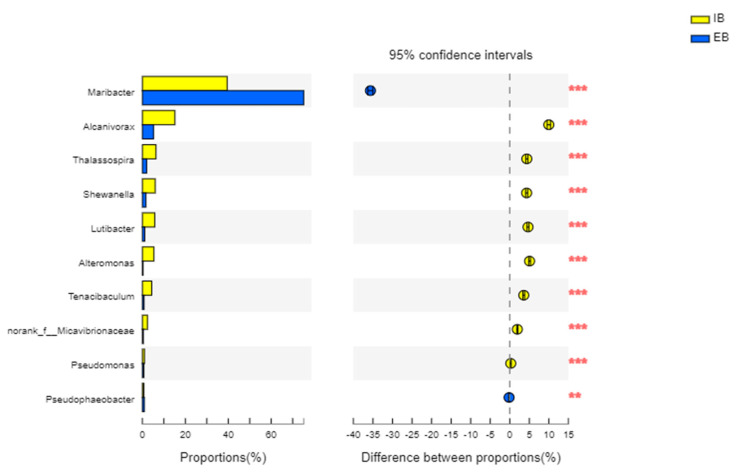
Fisher’s exact test bar plot at the genus level for the immobilized petroleum–degrading bacteria beads at different areas of the bioreactor (water inlet site (IB); water outlet site (EB), shown in yellow and blue, respectively). Asterisks (*) on the right represent significant differences: **: *p*-value was between 0.001 and 0.01; ***: *p*-value was below 0.001.

**Figure 7 ijerph-19-11348-f007:**
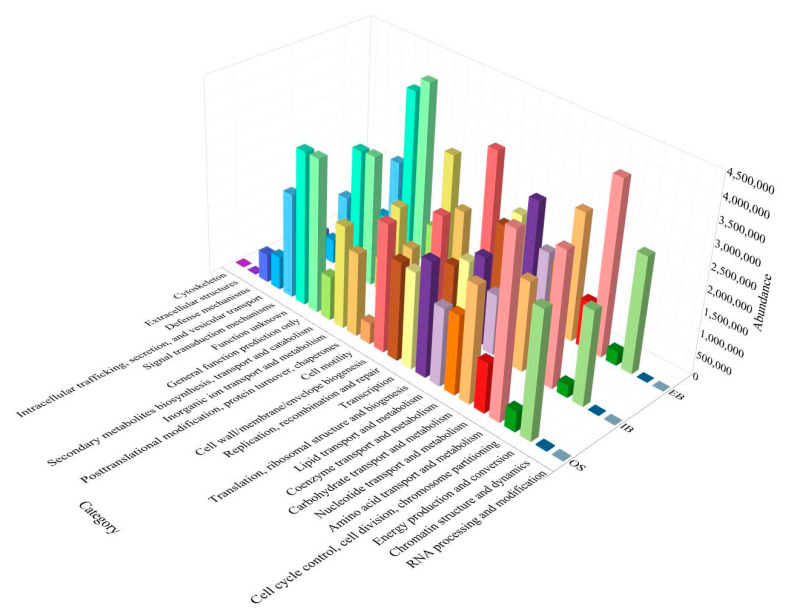
COG functional classification statistics for the immobilized petroleum–degrading bacteria beads in different areas (original seawater (OS); water inlet site (IB); water outlet site (EB)).

**Table 1 ijerph-19-11348-t001:** Details for the samples.

Sample	Name	Details
Original seawater	OS	Microbial sample of original seawater
Influent immobilized beads	IB	Microbial sample of 45-day immobilized petroleum–degrading bacteria beads in the influent site of the bioreactor
Effluent immobilized beads	EB	Microbial sample of 45-day immobilized petroleum–degrading bacteria beads in the effluent site of the bioreactor

**Table 2 ijerph-19-11348-t002:** Relationships between bonds and structure.

Bond (cm^−1^)	Related Structures
2800–3000	Cell membrane fatty acids
1500–1800	Cell proteins
1400–1500	Fatty acids
900–1200	Glycopeptides and phosphate groups of nucleic acid constituents

## Data Availability

Data can be requested from the corresponding author.

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
