# Peer review of "Study on the Changes in Immobilized Petroleum–Degrading Bacteria Beads in a Continuous Bioreactor Related to Physicochemical Performance, Degradation Ability, and Microbial Community"

_ijerph, 2022, doi:10.3390/ijerph191811348_

Round 1

Reviewer 1 Report (Previous Reviewer 2)

In this manuscript, "degradation" is repeatedly used in the experimental result, however, it is unclear whether diesel was degraded or removed by adsorption. The continuous experimental result with the beads without bacterial coating in this manuscript is required to figure out how much diesel was removed by adsorption onto the beads. Otherwise, it should be careful to use "degradation".

Author Response

Reviewer 2 Report (New Reviewer)

I found the revised article very interesting, since it reflects a problem in the process of bacterial degradation in a bioreactor, and that the authors thoroughly reviewed, finding a possible cause that the bioremediation process continues to be affected by the predominance of a genus of bacteria.

The background reflects precisely what I found in the review, that there is very little information on the study of the behavior of oil degradation processes promoted by bacteria in a bioreactor.

The article presents a clear wording and is consistent between what is stated in the introduction and the results obtained.

Its publication without further review is recommended, since the topic is relevant and presents originality in the work carried out.

Author Response

Reviewer 3 Report (New Reviewer)

Author Response

This manuscript is a resubmission of an earlier submission. The following is a list of the peer review reports and author responses from that submission.

Round 1

Reviewer 1 Report

Dear Authors, the issue of biodegradation of oil stains is very important and relevant in terms of environmental protection.

Your work, in my opinion, is very interesting and presents the next step in this research. It is written correctly, with introductory information, the purpose of the work and the principle of creating a biological reactor explained in detail. After the test information, conclusions are given at the end. This does not raise any objections.

I have this minor suggestion: the number of items on such an interesting topic is far too small.

However, I would suggest separating the results from the discussion for the sake of better understanding of the content presented. The authors in the discussion could have devoted more space to considerations of restoration of the condition of the biological reactor.  These are basically the most important elements of the text, and I think it is worth taking this step having such interesting laboratory data.

In general, I consider the text to be correctly written. My suggestions are not necessarily binding.

Reviewer 2 Report

This study present the microbial community change in the diesel and seawater mixture. Although the various aspects of microbial community have been analyzed, this manuscript failed to set the clear objective of the study as well as to bring the meaningful outcomes related with the petroleum degradation. Furthermore, it appears that the diesel was removed by the adsorption unto the bead rather than the biodegradation. The authors need to clarify the diesel removal mechanism in this study (for example, to report the experimental result with the beads without bacteria coating). 

Reviewer 3 Report

- The English makes it hard to understand the paper in many points. 

- Section 1.1. The analytical method to determine diesel concentration is not clear. I guess that the authors select a UV wavenumber in which a diesel component provides the maximum absorbance, but the units used in the equation are mL/L. Please, clarify. The same applies fo Figure 1. Why you did not measure COD (chemical oxygen demand), TOC (total organic carbon) or similar?

- Figure 7. Titles are not visible. Please increase font size.

- There is no discussion of the results. The manuscript looks like a technical report instead of scientific work.